# Mitochondrial genetic diversity, selection and recombination in a canine transmissible cancer

Andrea Strakova[1†], Máire Ní Leathlobhair[1†], Guo-Dong Wang[2], Ting-Ting Yin[2], Ilona Airikkala-Otter[3], Janice L Allen[4], Karen M Allum[5], Leontine Bansse-Issa[6], Jocelyn L Bisson[1], Artemio Castillo Domracheva[7], Karina F de Castro[8], Anne M Corrigan[9], Hugh R Cran[10], Jane T Crawford[11], Stephen M Cutter[4], Laura Delgadillo Keenan[12], Edward M Donelan[4], Ibikunle A Faramade[13], Erika Flores Reynoso[14], Eleni Fotopoulou[15], Skye N Fruean[16], Fanny Gallardo-Arrieta[17], Olga Glebova[18], Rodrigo F Häfelin Manrique[19], Joaquim JGP Henriques[20], Natalia Ignatenko[21], Debbie Koenig[5], Marta Lanza-Perea[9], Remo Lobetti[22], Adriana M Lopez Quintana[23], Thibault Losfelt[24], Gabriele Marino[25], Inigo Martincorena[26], Simón Martínez Castañeda[27], Mayra F Martínez-López[28], Michael Meyer[29], Berna Nakanwagi[30], Andrigo B De Nardi[8], Winifred Neunzig[5], Sally J Nixon[31], Marsden M Onsare[32], Antonio Ortega-Pacheco[33], Maria C Peleteiro[34], Ruth J Pye[31], John F Reece[35], Jose Rojas Gutierrez[36], Haleema Sadia[37], Sheila K Schmeling[38], Olga Shamanova[39], Richard K Ssuna[40], Audrey E Steenland-Smit[6], Alla Svitich[41], Ismail Thoya Ngoka[42], Bogdan A Vițălaru[43], Anna P de Vos[44], Johan P de Vos[45], Oliver Walkinton[31], David C Wedge[26], Alvaro S Wehrle-Martinez[46], Mirjam G van der Wel[47], Sophie AE Widdowson[40], Elizabeth P Murchison[1*]

*For correspondence: epm27@ cam.ac.uk

†These authors contributed equally to this work

Competing interests: The authors declare that no competing interests exist.

[1]Department of Veterinary Medicine, University of Cambridge, Cambridge, United Kingdom; [2]State Key Laboratory of Genetic Resources and Evolution, Yunnan Laboratory of Molecular Biology of Domestic Animals, Kunming Institute of Zoology, Chinese Academy of Sciences, Kunming, China; [3]International Training Center, Worldwide Veterinary Service, Aruvankadu, India; [4]Animal Management in Rural and Remote Indigenous Communities, Darwin, Australia; [5]World Vets, Fargo, United States; [6]Stichting Dierenbescherming Suriname, Paramaribo, Suriname; [7]Corozal Veterinary Hospital, University of Panama, Panama City, Panama; [8]Department of Clinical and Veterinary Surgery, São Paulo State University, São Paulo, Brazil; [9]St. George's University, True Blue, Grenada; [10]The Nakuru District Veterinary Scheme Ltd, Nakuru, Kenya; [11]Animal Medical Centre, Belize City, Belize; [12]Veterinary clinic Sr. Dog's, Guadalajara, Mexico; [13]National Veterinary Research Institute, Vom, Nigeria; [14]International Fund for Animal Welfare, Quintana Roo, Mexico; [15]Intermunicipal Stray Animals Care Centre, Perama, Greece; [16]Animal Protection Society of Samoa, Apia, Samoa; [17]Faculty of Veterinary Science, University of Zulia, Maracaibo, Venezuela; [18]Veterinary clinic BIOCONTROL, Moscow, Russia; [19]Veterinary clinic El Roble, Santiago de Chile, Chile; [20]OnevetGroup, Centro Veterinário Berna, Lisboa, Portugal; [21]Veterinary clinic Zoovetservis, Kiev, Ukraine; [22]Bryanston Veterinary Hospital, Bryanston, South Africa; [23]Veterinary Clinic Lopez Quintana, Maldonado, Uruguay; [24]Clinique Veterinaire de Grand Fond, Saint Gilles les Bains, France; [25]Department of Veterinary Sciences, University of Messina, Messina, Italy; [26]Wellcome Trust Sanger

Institute, Hinxton, United Kingdom; [27]Facultad de Medicina Veterinaria y Zootecnia, Universidad Autónoma del Estado de México, Toluca, Mexico; [28]School of Veterinary Medicine, Universidad de las Américas, Quito, Ecuador; [29]Touray & Meyer Vet Clinic, Serrekunda, Gambia; [30]The Kampala Veterinary Surgery, Kampala, Uganda; [31]Vets Beyond Borders, The Rocks, Australia; [32]Aniworld veterinary clinic, Kisumu, Kenya; [33]Faculty of Veterinary Medicine, Autonomous University of Yucatan, Merida, Mexico; [34]Interdisciplinary Centre of Research in Animal Health, Faculty of Veterinary Medicine, University of Lisbon, Lisboa, Portugal; [35]Help in Suffering, Jaipur, India; [36]Veterinary clinic Dr José Rojas, Los Andes, Chile; [37]University of Veterinary and Animal Sciences, Lahore, Pakistan; [38]Corozal Veterinary Clinic, Corozal Town, Belize; [39]Veterinary clinic Vetmaster, Ramenskoye, Russia; [40]Lilongwe Society for Protection and Care of Animals, Lilongwe, Malawi; [41]State Hospital of Veterinary Medicine, Dniprodzerzhynsk, Ukraine; [42]Kenya Society for Protection and Care of Animals, Nairobi, Kenya; [43]Clinical Sciences Department, Faculty of Veterinary Medicine, Bucharest, Romania; [44]Ladybrand Animal Clinic, Ladybrand, South Africa; [45]Veterinary Oncology Referral Centre De Ottenhorst, Terneuzen, Netherlands; [46]Faculty of Veterinary Sciences, National University of Asuncion, San Lorenzo, Paraguay; [47]Animal Anti Cruelty League, Port Elizabeth, South Africa

**Abstract** Canine transmissible venereal tumour (CTVT) is a clonally transmissible cancer that originated approximately 11,000 years ago and affects dogs worldwide. Despite the clonal origin of the CTVT nuclear genome, CTVT mitochondrial genomes (mtDNAs) have been acquired by periodic capture from transient hosts. We sequenced 449 complete mtDNAs from a global population of CTVTs, and show that mtDNA horizontal transfer has occurred at least five times, delineating five tumour clades whose distributions track two millennia of dog global migration. Negative selection has operated to prevent accumulation of deleterious mutations in captured mtDNA, and recombination has caused occasional mtDNA re-assortment. These findings implicate functional mtDNA as a driver of CTVT global metastatic spread, further highlighting the important role of mtDNA in cancer evolution.

## Introduction

The canine transmissible venereal tumour (CTVT) is a transmissible cancer that is contagious between dogs via the transfer of living cancer cells during coitus. The disease usually manifests as localised tumours involving the genital mucosa in both male and female domestic dogs. CTVT first arose from the somatic cells of an individual dog that lived approximately 11,000 years ago; it subsequently survived beyond the death of this original animal by metastasising to new hosts (*Murgia et al., 2006*; *Rebbeck et al., 2009*; *Murchison et al., 2014*; *Decker et al., 2015*). CTVT is found in dog populations worldwide, and is the oldest and most prolific cancer lineage known in nature (*Murchison et al., 2014*; *Strakova and Murchison, 2014*; *Strakova and Murchison, 2015*). The clonal evolution of CTVT renders this lineage a unique genetic tag with which to trace historical global dispersals of dogs together with their human companions. Furthermore, the extreme longevity of this lineage, its serial colonisation of genetically distinct allogeneic hosts and its occasional uptake of host mitochondrial DNA (mtDNA) by horizontal transfer (*Rebbeck et al., 2011*), provide opportunities to probe genetic vulnerabilities in cancer and to identify novel host-tumour interactions. We analysed 449 complete mtDNAs in CTVT and used these to investigate the frequency and timing of mtDNA horizontal transfer in this lineage; furthermore, we assessed the contribution of selection to CTVT mtDNA evolution and searched for evidence of mtDNA recombination.

**eLife digest** A unique cancer called canine transmissible venereal tumour (CTVT) causes ugly tumours to form on the genitals of dogs. Unlike most other cancers, CTVT is contagious: the cancer cells can be directly transferred from one dog to another when they mate. The disease originated from the cancer cells of one individual dog that lived approximately 11,000 years ago. CTVT now affects dogs all over the world, which makes it the oldest and most widespread cancer known in nature.

Like healthy cells, cancer cells contain compartments known as mitochondria that produce the chemical energy needed to power vital processes. Inside the mitochondria, there is some DNA that encodes the proteins that mitochondria need to perform this role. Changes (or mutations) to this mitochondrial DNA (mtDNA) may stop the mitochondria from working properly. CTVT cells have previously been found to occasionally capture mtDNA from normal dog cells, which suggests that replenishing their mtDNA may help promote CTVT cell growth. Furthermore, these captured mtDNAs act as genetic "flags" that can help trace the spread of the disease.

Here, Strakova, Ní Leathlobhair et al. analysed the mtDNA in CTVT tumours collected from over 400 dogs in 39 countries. The analysis shows that CTVT cells have captured mtDNA from normal dog cells on at least five occasions. Over the last 2,000 years, the disease appears to have spread rapidly around the world, perhaps transported by dogs travelling on ships along historic trade routes. CTVT may have only reached the Americas within the last 500 years, possibly carried there by dogs brought by Europeans. Likewise, CTVT probably only came to Australia after European contact.

The experiments also revealed that the most damaging types of mutations were absent from the mtDNA of CTVT, which suggests that fully functioning mitochondria play an important role in CTVT. Unexpectedly, Strakova, Ní Leathlobhair et al. found evidence that certain sections of mtDNA in some CTVT cells have been exchanged, or shuffled, with the mtDNA captured from normal dog cells. This type of "recombination" is not usually thought to occur in mtDNA, and has not previously been detected in cancer. Future studies will determine if this process is widespread in other types of cancer, including in humans.

## Results and discussion

To investigate the global CTVT population structure and estimate the frequency and timing of mtDNA horizontal transfer, we performed low-coverage whole genome sequencing (~0.3X whole genome coverage) on 449 CTVT tumours and 338 matched hosts collected from 39 countries across six continents (Materials and methods) (*Figure 1—figure supplement 1*, *Supplementary file 1*). MtDNA was sequenced at ~70X coverage, indicating that each CTVT cell carries approximately 470 mtDNA copies (*Figure 1—figure supplement 2*, *Supplementary file 2*, Materials and methods). CTVT was confirmed by identification of a characteristic rearrangement involving a long interspersed nuclear element (LINE) near the *MYC* locus (*Katzir et al., 1985*; *1987*) (*Supplementary file 3*).

We identified 1005 single point substitution variants and 27 short insertions and deletions (indels) in the CTVT mtDNA population (*Supplementary files 4*, *5*, *6*, *7*, *12*, *13*). CTVT mtDNA somatic substitution mutations (see Materials and methods) had the characteristic profile that is observed in human cancers, dominated by C>T and T>C mutations showing a striking strand bias (*Ju et al., 2014*) (*Figure 2—figure supplement 1*). This mutational process is probably replication-coupled, and mutations associated with this process appear to accumulate at a roughly constant rate in human cancers (*Ju et al., 2014*). A maximum likelihood phylogenetic tree constructed with mtDNA sequences from CTVT, matched hosts and 252 additional dogs (see *Supplementary file 8*) revealed that CTVT mtDNAs cluster in five distinct groups within dog mtDNA haplogroup A1 (*Figure 1A*, *Figure 1—figure supplement 3*, *Figure 1—source data 1*). These data suggest that CTVT mtDNAs have at least five independent origins, demarcating five groups that we have named CTVT clades 1 to 5.

Although CTVT originated about 11,000 years ago, whole genome sequences of two CTVT tumours derived from clades 1 and 2 indicated that these two clades shared a common ancestor

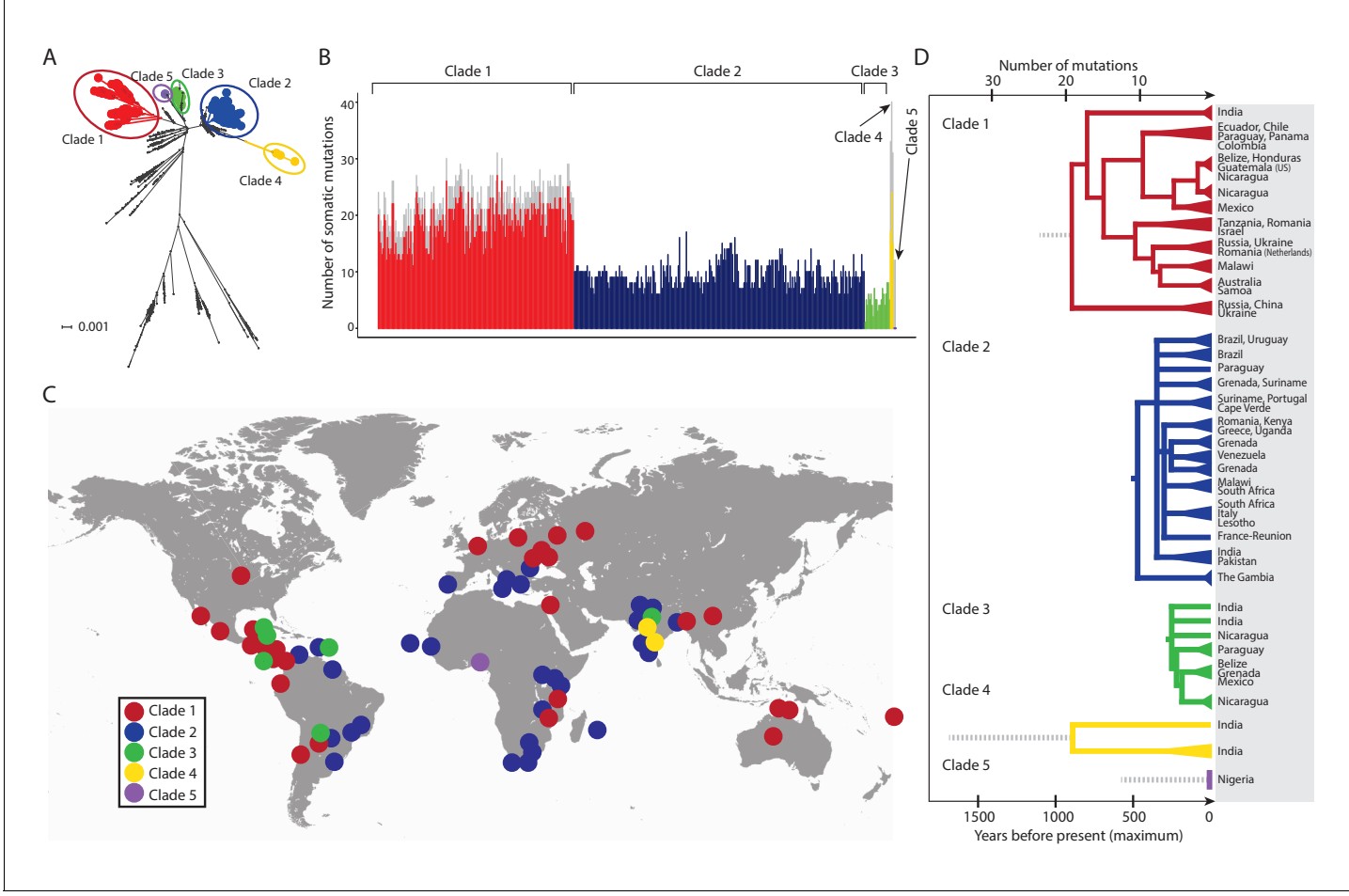

**Figure 1.** CTVT has acquired mtDNA by horizontal transfer at least five times. (**A**) Maximum likelihood phylogenetic tree constructed with complete mtDNA sequences from 449 CTVT tumours and 590 dogs. Coloured and black dots represent CTVT and dog mtDNA respectively. Scale bar indicates base substitutions per site. (**B**) Number of somatic substitution mutations per CTVT tumour. Coloured bars indicate somatic mutations acquired by each tumour since mtDNA capture. Grey bars indicate substitutions absent from normal dog mtDNA haplotypes but common to all tumours within a clade; thus the early somatic or rare germline status of these variants is unknown. (**C**) Geographical distribution of clades. Coloured dots represent locations from which one or more CTVT tumours were collected. (**D**) Simplified representation of maximum likelihood phylogenetic trees for each clade. Trees illustrate nodes with bootstrap support >60, and shaded triangles represent coalescence of individual branches within each group. Two tumours were collected in the United States and the Netherlands respectively from dogs imported from Guatemala and Romania. Discontinuous grey lines represent contributions of substitutions absent from normal dog mtDNA haplotypes but common to all tumours within a clade. Assuming a constant accumulation of mutations within and between clades, approximate number of somatic mutations and estimated timing is shown. Maximum likelihood trees upon which these representations are based are found in *Figure 1—source data 2*.

The following source data and figure supplements are available for figure 1:

**Source data 1.** Maximum likelihood phylogenetic tree of CTVT mtDNA.

**Source data 2.** Maximum likelihood phylogenetic trees for CTVT clades 1 to 5.

**Figure supplement 1.** Geographical locations and mtDNA clades for CTVT tumours and hosts.

**Figure supplement 2.** mtDNA copy number in CTVT.

**Figure supplement 3.** CTVT mtDNA clades 1 to 5 all arose from dog mtDNA clade A.

**Figure supplement 4.** Reconstructed donor haplotypes for CTVT mtDNA clades 1 to 5.

*Figure 1 continued on next page*

*Figure 1 continued*

**Figure supplement 5.** Sequence contribution of nuclear-encoded mtDNA (NuMTs).

approximately 460 years ago (*Murchison et al., 2014*). We investigated the relative time since each CTVT mtDNA horizontal transfer event by estimating the number of mtDNA somatic mutations acquired by each clade since mtDNA capture (*Figure 1B*). This analysis revealed that clade 1 mtDNA carry more than double the number of mtDNA somatic mutations (22.5 mutations average) compared with clade 2 mtDNA (9.4 mutations average). By inferring that the clade 2 mtDNA horizontal transfer event occurred no more than 460 years ago, this analysis suggests a maximum time since mtDNA uptake of 1097 years for clade 1, 244 years for clade 3, 1690 years for clade 4 and 585 years for clade 5, assuming a constant somatic accumulation of mutations in CTVT mtDNA (Materials and methods). Importantly, two additional mutation rate estimates, derived using human data (*Ju et al., 2014*), suggested similar timing for CTVT clade origins (Materials and methods, *Supplementary file 9*). Thus, this analysis suggests that the original mtDNA, that was present in the founder dog that first spawned CTVT, is not detectable in tumours that we have analysed, and indicates that CTVT cells have captured mtDNA from transient hosts at least five times within the last two thousand years.

The geographic distribution and phylogenies of the five CTVT clades reveal the dynamic recent history of the CTVT lineage (*Figure 1C*, *Figure 1D*, *Figure 1—figure supplement 1* and *Figure 1—source data 2*). Clades 1 and 2, which occur most frequently in the CTVT population that we analysed, both have a global distribution. Tumours that diverged early in the clade 1 lineage occur in Russia, Ukraine, China and India, suggesting an Old World origin for this clade (*Figure 1D*). Clade 1 tumours in Central and South America share a single common ancestor that probably existed no more than 511 years ago, suggesting introduction of CTVT to the Americas with colonial contact; similarly, our data suggest a single introduction of CTVT to Australia after European arrival (maximum 116 years ago) (*Figure 1D*, *Supplementary file 9*, see Materials and methods). The distribution pattern and timing of clade 2 suggest that this clade may have been transported between continents via trans-Atlantic and Indian Ocean trade routes (*Figure 1C and 1D*). The more recent clade 3 lineage was found in Central and South America and India, and the less frequent clades 4 and 5 occurred only in India and Nigeria respectively (*Figure 1C and 1D*). The extensive and recent global expansion detected in the CTVT lineage is consistent with signals of widespread admixture observed in worldwide populations of domestic dogs (*Shannon et al., 2015*), highlighting the extent to which canine companions accompanied human travellers on their global explorations.

Most somatic mutations in cancer are believed to be selectively neutral, and there is little evidence in human cancers for negative selection operating to safeguard essential cellular processes (*Stratton et al., 2009*). We searched for evidence of mtDNA functionality in CTVT cells by examining CTVT mtDNA for signals of negative selection. If present, negative selection would be expected to operate on mtDNA to prevent homoplasmy of deleterious mutations. Consistent with this prediction, the variant allele fraction (VAF) of nonsense substitutions and frameshift indels was significantly lower than VAF for other substitutions and indels (*Figure 2A and B*, p=0.00019 and p=3.03x10$^{-05}$ respectively, two-sample Kolmogorov-Smirnov test). Furthermore, dN/dS for somatic mtDNA mutations in CTVT showed significant deviation from neutrality both for nonsense (0.187, p=1.02x10$^{-07}$) and missense (0.748, p=4.18x10$^{-03}$) mutations (*Figure 2C*). Together with evidence of reduced VAF for truncating mtDNA mutations in human cancers (*Ju et al., 2014*; *Stewart et al., 2015*), these findings provide evidence for the activity of negative selection operating to preserve mtDNA function in CTVT and indicate that, at least in some cancers, functional mtDNA contributes to driving cancer.

MtDNA is usually assumed to be clonally inherited and recombinationally inert. However, mtDNA recombination has been directly observed in various eukaryotes (*Lunt and Hyman, 1997*; *Hoarau et al., 2002*; *Ladoukakis and Zouros, 2001*; *Gantenbein et al., 2005*; *Ujvari et al., 2007*; *Bergthorsson et al., 2003*) and has been proposed as a mechanism for mtDNA repair (*Thyagarajan et al., 1996*). Recombination of maternal and paternal mtDNA haplotypes has been observed in rare cases of human biparental mtDNA inheritance (*Kraytsberg et al., 2004*;

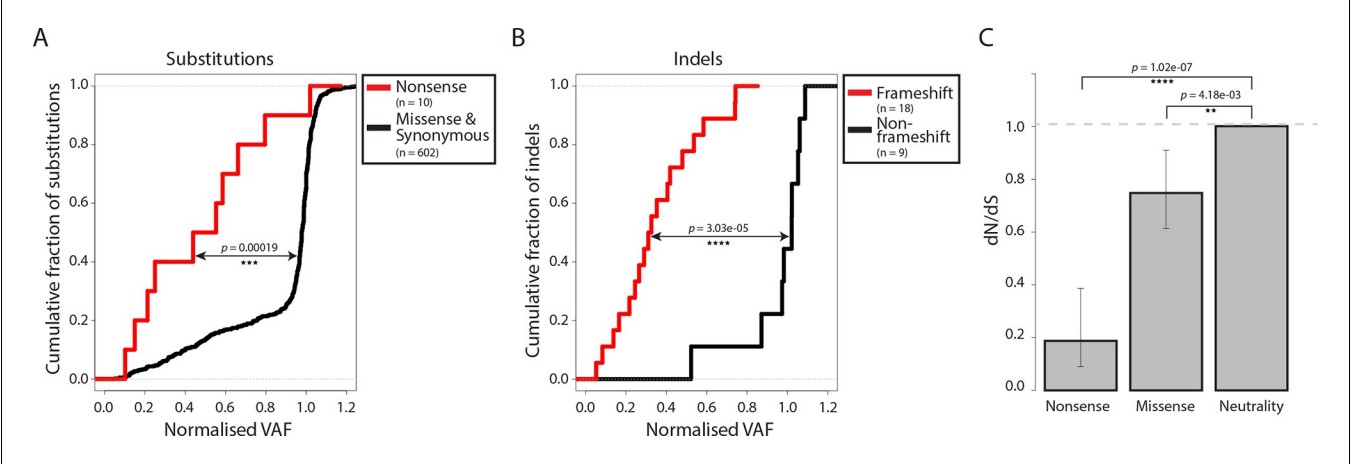

**Figure 2.** Negative selection operates to prevent the accumulation of gene-disrupting mutations in CTVT. Cumulative distribution functions for variant allele fraction (VAF) for gene-disrupting (**A**) substitutions and (**B**) indels. *P*-values were calculated using two-sample Kolmogorov-Smirnov tests. (**C**) dN/dS for somatic nonsense and missense substitutions. *P*-values were calculated using a likelihood ratio test with parameters estimated using a Poisson model. Error bars indicate 95 percent confidence intervals.

The following figure supplement is available for figure 2:

**Figure supplement 1.** CTVT mtDNA somatic mutation spectrum.

*Zsurka et al., 2005*), and mtDNA recombination activity is present in human cell extracts (*Thyagarajan et al., 1996*). However, mtDNA recombination has not, to our knowledge, been previously detected in cancer. Given the possibility for coexistence of two distinct mtDNA haplotypes in CTVT cells, we searched for evidence of mtDNA recombination in CTVT using recombination-detection algorithms 3seq and SiScan (*Boni et al., 2007*; *Gibbs et al., 2000*). Remarkably, these algorithms detected significant evidence for mtDNA recombination in CTVT clade 1, detecting recombination breakpoints at around MT:5430 and MT:16176. Maximum likelihood phylogenetic trees constructed using segments MT:1–5429 and MT:5430–16176 derived from clade 1 mtDNA produced distinct topologies (*Figure 3A*, *Figure 3—source data 1*). Further inspection of clade 1 mtDNA haplotypes suggested that recombination replaced MT:1–5429 in a clade 1 mtDNA haplotype that diverged from Central American clade 1 CTVTs and that subsequently colonised areas of South and Central America (Chile, Colombia, Ecuador, Panama, Paraguay) (*Figure 3B*).

These data provide evidence of an mtDNA recombination event in an ancestral CTVT lineage. We searched for evidence of more recent mtDNA recombination by examining outliers on CTVT mtDNA phylogenetic trees (*Figure 1—source data 2*, *Supplementary file 10*). This analysis identified 559T, a CTVT tumour derived from a male dog in Nicaragua (*Figure 3C*). Further investigation of mtDNA in 559T revealed what appeared to be a CTVT clade 1 mtDNA haplotype (CTVT_1B2b1_29) superimposed upon a dog mtDNA haplotype (A1d1a_1), neither of which resembled the mtDNA haplotype found in normal tissues from this host dog, 559H (B1_1 haplotype). Phasing of mtDNA variants in 559T using long sequence reads indicated the presence of at least three distinct mtDNA haplotypes in this tumour, each representing a recombination product apparently derived from mtDNA haplotypes CTVT_1B2b1_29 and A1d1a_1 (*Figure 3C*). These data suggest that a tumour antecedent of 559T captured haplotype A1d1a_1 mtDNA from its host. Recombination was initiated between mtDNA haplotypes CTVT_1B2b1_29 and A1d1a_1, and cells containing these recombination products were passed to host 559H. Alternatively it is possible that 559H received a mixture of both normal and CTVT cells from its CTVT donor animal, and mtDNA capture and recombination occurred within 559H. It must also be mentioned that the A1d1a_1 haplotype resembles the CTVT clade 3 donor haplotype (*Figure 1—figure supplement 4*); thus we cannot exclude the possibility that the

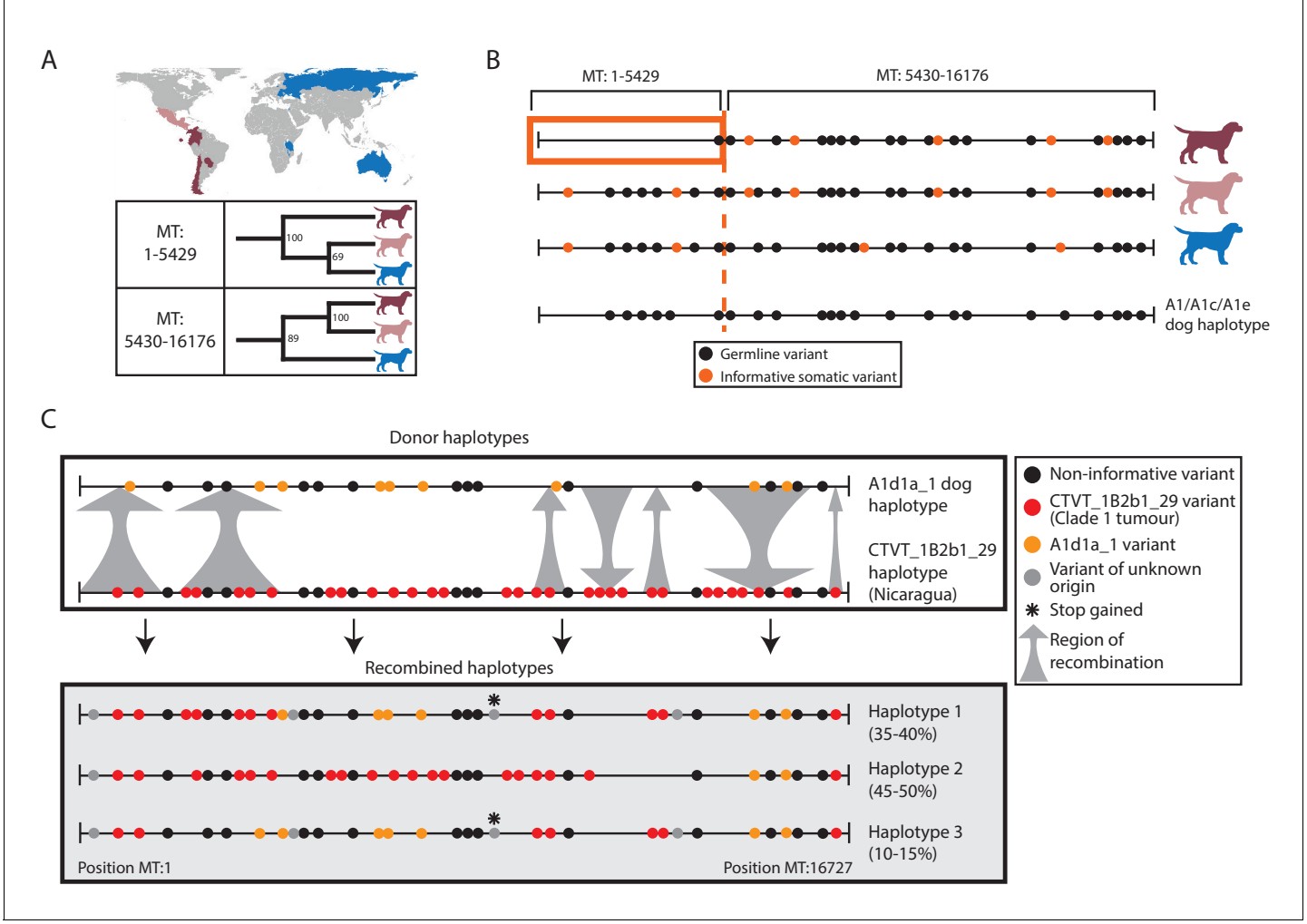

**Figure 3.** Ancient and modern mtDNA recombination in CTVT. (**A**) Maximum likelihood phylogenetic trees constructed using segments MT:1–5429 and MT:5430–16176 from clade 1 CTVT mtDNAs. Three clade 1 mtDNA haplotype groups are represented by coloured dog silhouettes, and their geographical distributions are colour-coded on the map. Bootstrap values were calculated from 100 iterations. Maximum likelihood trees upon which these representations are based are found in *Figure 3—source data 1*. (**B**) Simplified haplotype diagrams for clade 1 CTVT mtDNAs derived from groups shown in (**A**). Germline variants were present in the donor mtDNA that founded clade 1, represented by the A1/A1c/A1e dog haplotype (see *Figure 1—figure supplement 4*). Region putatively replaced by recombination is outlined with orange box. (**C**) Recombination detected in tumour 559T (Nicaragua). The estimated per cent contribution of each recombined haplotype to the mtDNA population within 559T CTVT cells is shown, and grey arrows indicate likely sites of recombination.

The following source data is available for figure 3:

**Source data 1.** Ancient mtDNA recombination in CTVT clade 1.

recombination that we observe in 559T involved horizontal transfer between clade 1 and clade 3 CTVT tumours that occurred within the same animal.

Our analysis provides evidence for occasional mtDNA recombination activity in CTVT cells. The mechanism whereby distinct mtDNA molecules are able to interact within the cell, and the nature of the signals that trigger onset of mtDNA recombination are not clear. Further analysis will determine if DNA damage signalling is involved, and it is interesting to observe that a truncating nonsense mutation in *COX3* was found in some 559T haplotypes (*Figure 3C*). Although we could not find evidence of mtDNA recombination in CTVT beyond those described, we cannot exclude the possibility that recombination is more widespread in CTVT mtDNA than detected. It is possible, therefore, that our

phylogenetic, mutation rate and selection analyses (*Figure 1*, *2*) have been influenced by an undetected recombination signal. However, the presence in all (non-recombining) CTVT mtDNAs of a set of clade-specific markers (*Figure 1—figure supplement 4*), the absence (beyond 559T) of distinctive phylogenetic outliers (*Figure 1—source data 2*), the very low frequency of back-mutation (*Supplementary file 10*), the strong somatic signal identified in the CTVT mtDNA mutational spectrum (*Figure 2—figure supplement 1*), and the failure of recombination-detection algorithms to detect further recombination, suggest that, if such a signal is present, it is at a low level.

CTVT is the world's oldest known cancer whose metastatic spread through its global host population provides unique insights into evolutionary processes operating in cancer. Our analysis of CTVT mtDNA has illuminated five mtDNA horizontal transfer events which trace two millennia of CTVT global spread. Negative selection has operated on CTVT to maintain mtDNA integrity at the level of nonsense and missense mutations, and occasional mtDNA recombination has occurred, possibly to repair damaged mtDNA. Evidence of negative selection demonstrates that maintenance of functional mtDNA is important for the biology of CTVT; and the observation of multiple mtDNA horizontal transfer events further supports the possibility that mtDNA capture from hosts is a positively selected adaptive mechanism (*Rebbeck et al., 2011*; *Tan et al., 2015*; *Spees et al., 2006*). This study highlights the important role of functional mtDNA in cancer and reveals unexpected biological mechanisms that have operated in an ancient mammalian somatic cell lineage.

## Materials and methods

### Sample collection and DNA extraction

This study was approved by the Department of Veterinary Medicine, University of Cambridge, Ethics and Welfare Committee (reference number CR174). Tumour and host (gonad, skin, blood or liver) tissue samples were collected into RNAlater solution and stored at 4°C until processing. Genomic DNA was extracted using the Qiagen DNeasy Blood and Tissue extraction kit. Sample information is presented in *Supplementary file 1*.

### Confirmation of canine transmissible venereal tumour (CTVT) diagnosis

Quantitative PCR (qPCR) assays were performed to confirm CTVT diagnosis (*Supplementary file 3*) by detection of the CTVT-specific *LINE-MYC* genomic rearrangement (*Murgia et al., 2006*; *Rebbeck et al., 2009*; *Murchison et al., 2014*; *Katzir et al., 1985*; *1987*). Each qPCR was performed in triplicate with SYBR Select Master Mix (Life Technologies, Carlsbad, CA) using an Applied Biosystems 7900HT Fast Real-Time PCR system instrument (Applied Biosystems, Foster City, CA) with conditions and primers specified below.

| Primer | | Sequence |
|---|---|---|
| *LINE-MYC* primers (obtained from [*Rebbeck, 2007*]) | Forward | AGG GTT TCC CAT CCT TTA ACA TT |
| | Reverse | AGA TAA GAA GCT TTT GCA CAG CAA |
| *ACTB* primers | Forward | CTC CAT CAT GAA GTG TGA CGT TG |
| | Reverse | CGA TGA TCT GAT CCT CAT TGT GC |

| qPCR master mix reagents | Volume per reaction (µl) |
|---|---|
| SYBR Green Mix | 10 |
| Primers (5 µM/primer) | 2.4 |
| DNA (20 ng/µl) | 0.5 |
| Water | 7.1 |
| Total volume | 20 |

| Stage of qPCR amplification | Temperature (°C) | Time (s) |
|---|---|---|
| Initial denaturation | 95 | 600 |
| 40 cycles | 95 | 15 |
|  | 60 | 60 |
| Final dissociation | 95 | 15 |

Standard curves were constructed for each primer set using CTVT tumour 29T1 as reference. Relative DNA input was calculated using standard curves as follows: (Ct = m(log10(iA)) + b), with each of the parameters defined as follows: Ct = threshold cycle, m = slope of the standard curve, iA = input amount, b = y-intercept of the standard curve. Relative DNA input for *LINE-MYC* was then normalised to *ACTB*. *LINE-MYC* and *ACTB* are present in three and two copies respectively in 24T and 79T CTVT tumours (*Murchison et al., 2014*); however, it is possible that copy number at these loci differs between tumours in the current dataset.

## DNA sequencing

Whole genome sequencing libraries with insert size 100 to 400 base pairs (bp) were constructed using standard methods according to manufacturer's instructions and sequenced with 75bp paired end reads on an Illumina HiSeq2000 instrument (Illumina, San Diego, CA) to an average whole genome depth of 0.3X; average mitochondrial DNA (mtDNA) coverage was ~70X. Reads were aligned with the CanFam3.1 dog reference genome (*Lindblad-Toh et al., 2005*) (http://www.ensembl.org/Canis_familiaris/Info/Index) using the BWA alignment tool (*Li and Durbin, 2009*). To calculate the mitochondrial copy number, we used the following equation: (mtCOV/nuclCOV)*P, where mtCOV = average coverage across the mitochondria, nuclCOV = average coverage across the nuclear genome and P = ploidy. The ploidy used in our calculations was 2 for both CTVT tumours and CTVT hosts (*Murchison et al., 2014*). Host and tumour samples with average MT coverage >300X were excluded from the copy number calculations (see *Supplementary file 2A*).

Samples 1380T and 1381T were sequenced separately, based on the methods described in Pang et al (*Pang et al., 2009*). Complete mitochondrial genomes were amplified using the primers listed in Pang et al (*Pang et al., 2009*) with a number of additional primers listed below. The PCR conditions are specified below.

| PCR master mix reagents | Volume per reaction (µl) |
|---|---|
| 1 X PCR LATaq buffer | 2.5 |
| Primer forward and reverse (10 µM) | 1.2 (each) |
| DNA (100-200 ng/µl) | 1.2 |
| LATaq DNA polymerase | 0.25 |
| 1X dNTP (10 mM) | 4 |
| Water | 14.65 |
| Total volume | 25 |

| Stage of PCR amplification | Temperature (°C) | Time (s) |
|---|---|---|
| Initial denaturation | 94 | 300 |
| 12 cycles (touchdown PCR program, reduce 1°C each cycle) | 94 | 60 |
|  | 61–50 | 60 |
|  | 74 | 90 |

*Continued on next page*

*Continued*

| Stage of PCR amplification | Temperature (°C) | Time (s) |
|---|---|---|
| 25 cycles | 94 | 60 |
| | 52 | 60 |
| | 74 | 90 |
| Final extension | 74 | 420 |

| Primer name | Sequence (5'–3') |
|---|---|
| D0132 | ACC GTA AGG GAA TGA TGA A |
| D0136 | TGT AAG TGG TCG TAG AGG TTC |
| D0141 | AGG CGG ACT AAA TCA AAC TCA |
| D0146 | GGG GTA TCT AAT CCC AGT TT |
| D0149 | AAG TTT GGT AGC ACG AAG AT |

The PCR products were purified using a 1.0% agarose gel and sequenced on a 3730xl DNA analyser (Applied Biosystems) with a Big Dye Terminator v3.1 Sequencing Kit (Applied Biosystems). The sequenced fragments were assembled by Seqman (DNASTAR, Madison, WI) and the complete mitochondrial genomes were aligned with the CanFam3.1 dog mitochondrial reference genome (*Lindblad-Toh et al., 2005*).

## Nuclear copies of mtDNA (NuMT) analysis
Nuclear copies of mtDNA (NuMTs) are mtDNA fragments that have been incorporated into the nuclear genome. Over 150 NuMTs have been identified in the canine genome (*Verscheure et al., 2015*). Somatically acquired NuMTs have also been described in human cancer (*Ju et al., 2015*).

Given that our study design did not involve purification of cytoplasmic mtDNA genomes, we assessed the possibility that our mtDNA variant analysis has been influenced by NuMTs.

### NuMTs in CanFam3.1
We first assessed the potential contribution of NuMTs present within the CanFam3.1 assembly to our variant calling. We used *wgsim* (https://github.com/lh3/wgsim) to simulate sequence reads from CanFam3.1 (excluding the MT chromosome) to a coverage of 0.3X (i.e. the average nuclear genome coverage sequenced as part of this study). We then used BWA (*Li and Durbin, 2009*) to align the reads to the CanFam3.1 MT reference and used Samtools depth (*Li et al., 2009*; *Li, 2011*) to assess the MT genome coverage. Any MT genome coverage detected from this analysis would be expected to arise from NuMTs. The average MT genome coverage from this analysis was 0 (*Figure 1—figure supplement 5*), indicating that the NuMTs known to be present within CanFam3.1 are at insufficient copy number and/or are too divergent to map to the MT reference genome using the alignment parameters used in this study.

### Somatically acquired NuMTs
The analysis described above confirms that NuMTs that form part of the CanFam3.1 assembly have not impacted on the variant analysis performed in this study. However, it is possible that somatically acquired NuMTs that are not captured in the CanFam3.1 assembly could confound our variant analysis.

The following observations argue against the possibility that NuMT-derived variants have had a significant impact on our tumour variant calling:

- As CTVT is a clonal lineage, somatically acquired NuMT-derived variants would be expected to present as stable low-VAF variants across all tumours within a phylogenetic group. Variants with these features were not observed.

- The mutation spectrum that we observed in CTVT mtDNA has the distinctive profile characteristic of the known somatic mtDNA mutational process (*Figure 2—figure supplement 1* and *Ju et al., 2014*). As this mutational process is specific to cytoplasmic mtDNA, this observation suggests that the majority of variants within our set are of cytoplasmic origin.

## Substitution calling

### Extraction and filtering

Substitutions were called using CaVEMan (Cancer Variants through Expectation Maximisation), an in-house variant calling algorithm, as previously described (*Nik-Zainal et al., 2012*) (http://cancerit.github.io/CaVEMan/). As CaVEMan is designed for matched tumour-normal data, and CTVT tumours and normals are unmatched (i.e. they are different individuals), we used simulated reads derived from the reference genome as the 'normal', and called all substitution variants relative to this. A variant allele fraction (VAF value, i.e. number of reads supporting the substitution variant as a fraction of the total number of reads covering the substitution variant position) was reported for each substitution detected. The following list of in-built post-processing filters was used to improve the specificity and sensitivity of substitution calls:

- At least one third of mutant alleles must have base quality >25.
- Mean mapping quality of reads supporting a substitution variant call must be ≥21.
- Substitution variant calls supported only by the first or last 15bp of reads were discarded.
- Substitution variants were discarded if they occurred 10bp upstream or downstream of an unfiltered indel called in the same sample (as detected by the indel-detecting algorithm cgpPindel see 'Indel calling-Extraction and filtering'). The 10bp range was extended by the REP value for samples where an indel had been called with REP>0; REP represents the number of times the inserted/deleted base(s) occurs in the sequence directly 5' or 3' of the putative indel.
- Substitution variant calls were discarded if they occurred within region MT:16129-16430 inclusive; this is a simple repeat region as defined by the UCSC (http://genome.ucsc.edu/) table browser (Dog, CanFam3.1).
- If the reference allele was supported by at least one read on both strands (forward and reverse), then we required that the mutant allele should be supported by at least one read on both strands.

Substitutions in 1380T and 1381T were called using MEGA (Molecular Evolutionary Genetics Analysis) (*Tamura et al., 2013*).

### Post processing

#### Somatic substitutions in tumours with matched hosts

To remove tumour substitutions caused by host contamination, substitutions that were called in both tumour and matched host, but had VAF<0.9 in the tumour, were discarded. Substitutions with VAF>0.9 in both tumour and matched host were considered to be likely germline substitutions shared between host and tumour, and were retained. Low coverage hosts (defined as average coverage <20X, *Supplementary file 2A*) were additionally checked for evidence of substitutions at positions where substitutions were called in the corresponding tumour, and the substitution was discarded in the tumour if at least one read supporting the substitution was found in the low coverage host. All tumour substitutions with VAF<0.5 were discarded if the matched host was of low coverage. Low-level tumour-contaminated hosts were additionally checked for the presence of substitutions identified in other tumours (see *Supplementary file 4A*) and any substitutions arising due to contamination were discarded.

#### Somatic substitutions in tumours without matched hosts

VAF value was used to identify substitutions likely arising due to host contamination in tumours for which matched hosts were not available (see *Supplementary file 1* for tumours without matched hosts). We used VAF plots, which display VAF value versus genomic position, to identify the level of host contamination in each tumour. We then discarded any substitution below a VAF cutoff, specified uniquely for each tumour based on its estimated level of host contamination (for most tumours, the VAF cutoff was 0.5 or 0.6). If VAF plots did not show clear distinctions between tumour

substitutions and host contamination (i.e. host contamination was greater than ~40%; this category included 9 tumours), we identified likely tumour substitutions as those which were present in phylogenetically-related tumours. Remaining substitutions were removed if they were also found in normal dogs (see 'CTVT host and published dog genome germline substitutions list', *Supplementary file 4F*); those substitutions that were not found either in phylogenetically-related tumours or in normal dogs were kept as putative somatic substitutions (total of 11 substitutions).

## Germline substitutions in hosts

Substitutions in hosts were filtered using filters described in 'Substitution calling-Extraction and filtering'. Low coverage hosts (defined as average coverage <20X, *Supplementary file 2A*) were further checked for evidence of substitutions at positions where a substitution was called in the corresponding tumour as described in 'Somatic substitutions in tumours with matched hosts'.

Caution should be taken when considering substitution lists for low coverage hosts and hosts with regions of low coverage (see *Supplementary file 2A* for average coverage per sample and *Supplementary file 2B* for list of samples with low coverage mtDNA regions), as these may contain false negatives due to low coverage.

## Additional quality checks and validation

Additional quality filtering was performed in low coverage regions, regions with low-mapping quality, and regions containing variable-length polyC homopolymer tracts (*Fregel et al., 2015*). Substitutions that were subsequently completely excluded from the analysis as a part of this check are listed in the table below.

| Position | Base change | Justification for excluding substitution |
|---|---|---|
| 15493 | G>A | Inconsistently called due to low coverage and decreased mapping quality in this region |
| 15505 | T>C | Inconsistently called due to low coverage and decreased mapping quality in this region |
| 15632 | C>T | Frequently miscalled due to decreased mapping quality in this region |
| 15639 | T>A | Frequently miscalled due to decreased mapping quality in this region |
| 15639 | T>G | Frequently miscalled due to decreased mapping quality in this region |
| 15931 | A>G | Frequently miscalled due to its presence in the same position as a frequently miscalled indel (middle of a homopolymer tract) |
| 16672 | C>T | Frequently miscalled due to its proximity to a frequently miscalled indel (associated with a very long homopolymer tract) |
| 16705 | C>T | Frequently miscalled due to low coverage in this region |

Substitutions that were discarded due to proximity to an indel (see 'Substitution calling-Extraction and filtering' above) were visually inspected in Integrative Genomics Viewer (IGV) (*Robinson et al., 2011*; *Thorvaldsdottir et al., 2013*). A subset of these substitutions had substantial support and were rescued (listed below).

| Position | Base change |
|---|---|
| 381 | T>A |
| 1481 | T>C |
| 1683 | T>C |
| 2682 | G>A |
| 2683 | G>A |
| 3028 | A>C |
| 6629 | T>C |
| 6882 | A>G |

*Continued on next page*

*Continued*

| Position | Base change |
| --- | --- |
| 7014 | T>G |
| 8281 | T>C |
| 8368 | C>T |
| 8703 | G>A |
| 9825 | G>A |
| 9896 | T>C |
| 13708 | C>T |
| 14977 | T>C |
| 15524 | C>T |
| 15526 | C>T |
| 16660 | T>C |
| 16663 | C>T |
| 16671 | T>C |

## Host contamination

Host contamination levels in each tumour were estimated from VAF plots (see 'Somatic substitutions in tumours without matched hosts' above). Substitutions that were present in tumours but not in matched host were identified, and their average VAF used to estimate the proportion of tumour mtDNA (see *Supplementary file 2C* for estimated tumour cell fraction in each tumour). Substitution VAFs were normalised to take account of host contamination in *Figure 2A* and *Supplementary files 4B,C*. Tumour variants with normalised VAF<1 most likely represent heteroplasmic variants; however, we cannot exclude that these represent cellular subclones harbouring distinct homoplasmic mtDNA populations.

## Recurrent mutations and back mutations

Back mutations and recurrent mutations occurring in tumours were identified by inspecting positions of tumours carrying each substitution on phylogenetic trees (*Figure 1—source data 2*).

## Predicted functional consequences of substitutions

Variant effect predictor (VEP) (*McLaren et al., 2010*) was used to predict the functional consequences of single point substitutions, as annotated in *Supplementary file 6*.

## Extracting substitution variants from publicly available dog sequences

In order to enrich our panel of germline substitutions created from 338 CTVT hosts, we included an additional set of 252 publicly available complete dog mtDNA genomes (see *Supplementary file 8*). To extract substitution variants from available fasta files, sequences were aligned with the CanFam3.1 dog mitochondrial reference genome using Clustal Omega (*Sievers et al., 2011*). Alignment errors in the multiple sequence alignment, usually due to miscalls caused by closely mapped indels, were inspected manually and corrected to minimise gaps. Substitutions were extracted using snp-sites (https://github.com/sanger-pathogens/snp_sites). For those samples missing data in regions MT: 15510–15532 and MT: 16040–16550 we substituted the most likely substitution at polymorphic sites based on phylogenetic position. Our filtering rules were applied to the substitution set where applicable (see 'Substitution calling-Extraction and filtering' and 'Additional quality checks and validation') and substitutions called before MT position 48 or after MT position 16671 were excluded due to low coverage in these regions in our sequencing data. Substitutions represented by International Union of Pure and Applied Chemistry (IUPAC) codes R, Y, S, W, K or M where one of the two possible calls was the same

as the reference were changed to the base which was different to the reference. In cases where the IUPAC code represented >2 bases (B, D, H, V, or N), the reference base was substituted.

### Indel calling
## Extraction and filtering
Small insertions and deletions (indels) were extracted from the sequencing data using cgpPindel (https://github.com/cancerit/cgpPindel). The following list of in-built filters was used to improve the specificity and sensitivity of indel calls:

- Indels were required to have ≥3 supporting reads on either the forward or reverse strands or ≥2 supporting reads on both the forward and reverse strands
- Indel calls with at least 4 supporting pindel-mapped reads were required to have at least 1 supporting BWA-mapped read or, failing that, if REP= 0 (see 'Substitution calling-Extraction and filtering'), then at least one supporting pindel-mapped read on both strands.
- Indels called within the simple repeat region MT:16129–16430 inclusive were excluded.

Samples with very high coverage of the mitochondrial genome (24T-Dog, 24H-Dog, 1T-Dog, 2T-Dog, 3T-Dog, 4T-Dog, 4H-Dog, 498H-Dog, 432T-Dog, 455T1-Dog, 231T-Dog, 79H-Dog, 79T-Dog), see *Supplementary file 2A*, were excluded from this analysis due to frequent false positives, together with samples 1380T-Dog and 1381T-Dog.

## Post processing
## Somatic indels
Indels called in tumours that were also called in at least one host were discarded as possibly arising due to host contamination. Indels that were uniquely called in tumours without matched hosts were discarded, as we could not rule out the possibility that they were caused by host contamination. All remaining indels were visually validated using IGV (*Robinson et al., 2011*; *Thorvaldsdottir et al., 2013*). The indels listed in the table below were discarded from the analysis as miscalls. In total 27 somatic indels were included in the analysis (see *Supplementary file 5A*).

| Position | Indel | Sample | Justification for discarding indel |
|---|---|---|---|
| 9891 | TGATTTATCTCATAATTATCA> TATCTCATAATTATCATG | 324T2-Dog | Indel miscalled due to close proximity of two other indels in the same sample |
| 9910 | C>CAT | 401T-Dog | Indel miscalled due to presence of a substitution at the same position |

## Germline indels
The indels found on the host variant lists (*Supplementary files 5B* and *7B*, *Supplementary file 13*) include only those which were considered homoplasmic (VAF≥0.9) and which passed a visual validation performed using IGV (*Robinson et al., 2011*; *Thorvaldsdottir et al., 2013*).

## Recurrent indels
Recurrent indels occurring in tumours were identified by inspecting phylogenetic positions of tumours carrying each indel (*Figure 1—source data 2*).

## Variant Allele Fraction calculation for indels
Although primary indel calling was carried out using cgpPindel, indel allele fraction for wild type and mutant indels was calculated using vcfCommons (unpublished in-house software developed at the Wellcome Trust Sanger Institute; for additional information please contact cgp@sanger.ac.uk). The algorithm takes all mapped and unmapped reads in the region of an indel and performs an alignment of the reference and the predicted mutated path using Exonerate (*Slater and Birney, 2005*). The predicted mutated path is the reference with the change predicted by cgpPindel applied to it. Based on this alignment, reads are classified into 3 categories:

1. aligns to reference path

2. aligns to mutated path
3. ambiguous; a read sequence aligns to the reference and mutated path with identical scores which makes it impossible to determine the true path

Indel VAF values were normalised to take account of host contamination in *Figure 2B*.

## Predicted functional consequences of indels

Variant effect predictor (VEP) (*McLaren et al., 2010*) was used to predict the functional effects of indels, as listed in *Supplementary file 7*.

## Phylogenetic analyses

### Phylogenetic trees

Phylogenetic trees were constructed using a maximum likelihood (ML) method implemented in PhyML 3.0 (*Guindon et al., 2010*) using the General Time Reversible (GTR) + G + I nucleotide substitution model with transition/transversion (ts/tv) ratio, gamma distribution shape parameter and proportion of invariant sites estimated in PhyML for each dataset. Phylogenetic trees constructed using RAxML 8.2.6 with a rapid climbing algorithm and identical model specification were consistent with the phylogenetic trees obtained using PhyML. Prior to tree construction jModelTest2 (*Darriba et al., 2012*) (https://code.google.com/p/jmodeltest2) was used to determine the best fitting substitution model for the alignment, inferred using the Akaike Information Criteria (AIC). The tree topology was estimated using a combination of Nearest Neighbor Interchange (NNI) and Subtree Pruning and Regrafting (SPR) (*Hordijk and Gascuel, 2005*) algorithms. Trees were visualised using Dendrocope (*Huson and Scornavacca, 2012*). Bootstrap support values were obtained from 100 bootstrap replicates. Nodes with ≥60% support are labelled in *Figure 1—source data 2* and *Figure 3—source data 1*. The long branch-lengths in CTVT clade 4 were further investigated by visually validating sequence alignments involving these samples.

### Confirmation of individual horizontal transfer events

### Classification of tumour substitutions

CTVT tumour substitutions were classified as follows:

1) Tumour germline clade-defining substitutions (see *Supplementary file 4D* for the full list)

- Inferred to be present on the donor mtDNA haplotype which founded each of the five horizontal transfer events
- Present within the pool of substitutions in the current dog population (see *Supplementary file 4F*)
- Shared by all tumours within each clade derived from the original CTVT cell which received the horizontally transferred donated mtDNA (see *Figure 1—figure supplement 4*)

2) Tumour somatic substitutions (see *Supplementary file 4B* for the full list)

- Inferred to have arisen after the clade-defining horizontal transfer event
- Variable and phylogenetically informative within the set of tumours in one clade
- Any substitutions which were associated with inferred recombination event (present uniquely) in 559T were discarded (see 'Mitochondrial recombination analysis')

3) Tumour somatic substitutions - conservative list (see *Supplementary file 4C* for the full list)

- Inferred to have arisen after the clade-defining horizontal transfer event
- Variable and phylogenetically informative within the set of tumours in one clade
- Any substitutions which were classed as somatic but had the potential to be germline were excluded from this list. Exclusions were based on:
  - Inferred recombination events
  - Occurrence in clade 3; clade 3 tumours carry very few somatic mutations, and so the possibility that clade 3 tumours arose from several independent mtDNA horizontal transfer events cannot be excluded
  - Occurrence on ancestral trunks; these include trunks defining CTVT_1A, CTVT_1B1, CTVT_2A haplogroups; due to their ancestral divergence, it cannot be excluded that these haplotypes are derived from independent mtDNA horizontal transfer events

- ▪ Occurrence as a putative somatic mutation in tumours without hosts (see 'Somastic substitutions in tumours without matched hosts')
- This list was used for the analysis in *Figure 2A,C* and *Figure 2—figure supplement 1*.

4) Tumour potential somatic substitutions (see *Supplementary file 4E* for the full list)

- Substitutions present within all tumours within one clade but not represented in the set of 590 normal dogs analysed as part of this study (*Supplementary file 4F*)
- It cannot be determined if these substitutions are germline substitutions which were present on the mtDNA genome that founded each clade, or if they are early somatic substitutions that occurred after the clade-defining horizontal transfer event, but prior to the divergence of tumours analysed in our study.

## Definition of a CTVT clade

A CTVT clade is defined as a group of tumours arising from an individual horizontal transfer event. Each clade is defined with respect to the tumour substitution classification (see 'Classification of tumour substitutions') as follows:

1. CTVT tumours from the same clade cluster together on phylogenetic tree represented in *Figure 1A*.
2. CTVT tumours from the same clade arose from a single donor mtDNA haplotype and therefore share the same set of germline substitutions which was inferred to be originally present on the donor mtDNA haplotype (germline clade-defining substitutions, see *Figure 1—figure supplement 4*, 'Classification of tumour substitutions').
3. The reconstructed donor mtDNA haplotype for each clade has a phylogenetically closely related/identical haplotype in the current dog population (see *Figure 1—figure supplement 4*).

## Estimated timing of clade divergence

Average number of somatic mutations (i.e. mutations arising after the horizontal transfer event) within each clade was calculated for each of the five clades (see *Supplementary file 9*). For potential somatic mutations (shown in grey in *Figure 1B and D*), we were unable to determine whether they occurred before or after the horizontal transfer event, as they are present in all samples from the same clade, but not in the pool of germline substitutions (see *Supplementary file 4F*). As the number of somatic mutations influences the time of divergence, we included estimates both with and without potential somatic mutations in our analysis (see *Supplementary file 9*). Presence of mitochondrial recombination in haplogroup CTVT_1B2b2 (clade 1) was taken into account when calculating the average number of mutations per clade (see 'Mitochondrial recombination analysis'). The timing of clade divergence was estimated independently based on the following three methods, as explained below: timing based on nuclear DNA (*Murchison et al., 2014*; *Alexandrov et al., 2013*), timing based on number of cell divisions per homoplasmic mitochondrial mutation (*Ju et al., 2014*; *Cohen and Steel, 1972*), and timing based on number of mitochondrial mutations per year (*Ju et al., 2014*). All estimates of CTVT dates assume a constant rate of accumulation of somatic mtDNA mutations both within and between clades, including constant activity of selection (*Figure 2*).

### Timing based on nuclear DNA

Based on the most recent common ancestor of samples 24T (CTVT clade 1) and 79T (CTVT clade 2) which is estimated to have existed approximately 460 years ago (*Murchison et al., 2014*), we can assume that the maximum age of clade 2 (i.e. the more recent clade) is 460 years. Calibrated according to this dating (*Murchison et al., 2014*; *Alexandrov et al., 2013*), and assuming a constant rate of accumulation of mutations with time, the maximum number of CTVT somatic mtDNA mutations per year is 0.0205 (calculated using average number of mutations in clade 2 = 9.437). The calculated maximum time since origin of clades 1, 3, 4 and 5, is shown in *Figure 1D* and *Supplementary file 9*.

### Timing based on number of cell divisions

A study of mtDNA mutations in human cancers estimated that one homoplasmic somatic mtDNA mutation arises every ∼ 1000 cell generations in human cancers (*Ju et al., 2014*). An experimental

study estimated CTVT cell generation times to be 4 days for first stage tumours and >20 days for second stage tumours (*Cohen and Steel, 1972*). Using 4 days and 20 days as a minimum and maximum generation time, and assuming a constant mtDNA somatic mutation rate in CTVT, we estimated a minimum and maximum mutation rate of ~0.0183 and ~0.0913 mutations/year respectively. See *Supplementary file 9* for corresponding calculations of minimum and maximum time since clade origins.

## Timing based on number of mutations per year

A previous study correlated somatic mtDNA mutation accumulation in human cancers with patient age (*Ju et al., 2014*). This suggested an approximate rate of ~0.025 mutations per year. Assuming a similar rate in CTVT somatic mtDNA mutations and a constant CTVT somatic mtDNA mutation rate, we estimated time since mtDNA horizontal transfer events (*Supplementary file 9*).

## Haplotype analysis

### Haplotype nomenclature

### Host haplotypes

The host haplotype naming system was adapted from the cladistic canine mitochondrial DNA phylogeny nomenclature proposed by Fregel et al (*Fregel et al., 2015*). See *Supplementary files 1* and *11* for all host haplotype names. Corresponding CTVT normal dog host samples were assigned into one of the major clades (A, B, C, D, E and F). For subsequent levels, haplogroups were defined by specific diagnostic variants, as defined by Fregel et al (*Fregel et al., 2015*). A unique number, following an underscore after the haplogroup name, distinguished distinct haplotypes within each haplogroup. Haplogroup-defining variants 15632 C>T, 15639 T>A and 15639 T>G were excluded from our analysis ('Additional quality checks and validation') and therefore we were unable to distinguish between haplogroups A1c and A1e. Haplotypes which did not fit into any haplogroup were classified as 'unassigned'.

### Tumour haplotypes

We devised a CTVT mtDNA haplotype naming system adapted from the cladistic canine mtDNA phylogeny nomenclature proposed by Fregel et al (*Fregel et al., 2015*). See *Supplementary file 1* and *11* for all CTVT tumour haplotype names. Each CTVT haplotype has a prefix 'CTVT_', indicating a tumour haplotype. Distinct CTVT clades are numbered (CTVT_1, CTVT_2, CTVT_3, CTVT_4 and CTVT_5). For subsequent levels, hierarchical notation was used, where subsequent haplogroups were named by alternating letters and numbers and the maximum number of levels included in the hierarchical notation was five – i.e. 3 numbers and 2 letters (e.g. 1A1a1). The first letter is a Roman capital; subsequently-used letters are lower case Roman letters. Any haplogroups beyond the maximum number of levels were considered as a single subgroup, in which individual haplotypes were distinguished using a non-hierarchical numbering system - an underscore followed by a number (e.g. 1A1a1_1, 1A1a1_2, etc.). Underscores were only used to distinguish individual haplotypes after the haplotype has been assigned to all 5 hierarchical levels (e.g. 1A_1 does not exist, as this haplotype would be classified as 1A1 instead).

### Reconstructed donor haplotypes

A 'donor haplotype' was reconstructed for each of the clades, representing the inferred donor mtDNA haplotype in each horizontal transfer event, and was used to root the trees for each clade in *Figure 1—source data 2*. Donor haplotypes were reconstructed from the clade-defining germline substitutions and the clade-defining potential somatic substitutions (see 'Classifications of tumour substitutions' above) and are shown in *Figure 1—figure supplement 4*. The phylogenetically closest haplotype present in the current dog population is shown in the same figure.

## Mitochondrial recombination analysis

### Automated recombination analysis

The RDP4 package (*Martin et al., 2015*) was used to detect recombination events within the complete sample set (i.e. 449 CTVT tumours, 338 CTVT hosts and 252 additional dogs) using a Bonferroni corrected *p*-value cutoff of 0.05. Default parameters were used for the following programs

implemented within the RDP package: RDP (*Martin and Rybicki, 2000*), MaxChi (*Smith, 1992*), Chimaera (*Posada and Crandall, 2001*), 3Seq (*Boni et al., 2007*) and SiScan (*Gibbs et al., 2000*).

## Long read sequencing

A genomic library was created directly using 5µg of genomic DNA from sample 559T, not utilizing shearing or amplification techniques, as previously described (*Coupland et al., 2012*). The library was sequenced using two PacBio SMRT cells using the Pacific Biosciences RS sequencer (Pacific Biosciences, Menlo Park, CA). Each SMRT cell yielded ~1Gb of sequence data with mean read length 11,421bp and N50 read length 19,382bp. Average sequence coverage across the mitochondrial genome was 111.3X.

## PacBio data analysis

PacBio sequence reads aligning to mtDNA were viewed in SMRT view (Pacific Biosciences) as well as in Integrative Genomics Viewer (IGV) (*Robinson et al., 2011*; *Thorvaldsdottir et al., 2013*) and used to phase the mitochondrial substitutions previously called in 559T (Nicaragua). The three most common recombinant haplotypes were completely phased, as shown in *Figure 3C*. Additional haplotypes, which we were unable to phase completely and which were present at very low level (less than 5%), were also identified. Reads derived from 559H, the host of 559T (haplotype B1_1), were also identified. The reads used to phase the substitutions in the three most common haplotypes are shown in the table below:

**PacBio sequencing reads used for phasing of individual haplotypes**

| 559T haplotype 1 | 559T haplotype 2 | 559T haplotype 3 |
|---|---|---|
| m150625_205815_00127_c10080924255000000 1823177310081544_ s1_p0/80046 | m150623_234631_00127_c10078774255000000 1823173008251557_ s1_p0/328 | m150625_205815_00127_c10080924255000000 1823177310081544_ s1_p0/64645 |
| m150625_205815_00127_c10080924255000000 1823177310081544_ s1_p0/50141 | m150623_234631_00127_c10078774255000000 1823173008251557_ s1_p0/89158 | m150625_205815_00127_c10080924255000000 1823177310081544_ s1_p0/29160 |
| m150623_234631_00127_c10078774255000000 1823173008251557_ s1_p0/32367 | m150625_205815_00127_c10080924255000000 1823177310081544_ s1_p0/45213 | Many reads in the region between 9790-16627 |
| m150623_234631_00127_c10078774255000000 1823173008251557_ s1_p0/145458 | | |

## Mutation spectrum

The two strands of the mtDNA are known as the heavy and light strands, and the light strand is the reference strand in CanFam3.1. Each mutation on the conservative somatic list (n=835, 'Classification of tumour substitutions', *Supplementary file 4C*) was classified as one of six possible substitutions in the pyrimidine context (C>A, C>G, C>T, T>A, T>C, T>G) and assigned to a strand relative to the reference (i.e. pyrimidine mutations (i.e. C>, T>) with respect to the reference were defined as light strand mutations; purine mutations (i.e. A>, G>) with respect to the reference were defined as heavy strand mutations). The immediate 5' and 3' sequence contexts for each CTVT mutation was extracted from the dog mitochondrial reference genome for mutations on the heavy and light strands, yielding a maximum of 96 mutation types (6 possible substitutions x 4 possible 5' bases x 4 possible 3' bases).

The number of observations of each substitution type was normalised to the triplet frequency extracted from the canine mitochondrial genome. The following example illustrates how to calculate the observed/expected ratio for T[C>T]G occurring on the heavy strand. We observed a total of 835 substitutions occurring across the MT genome; given that the TCG triplet is observed 117 times in

the dog mitochondrial reference genome heavy strand, the frequency of TCG triplets occurring in the dog mtDNA heavy strand is 117/16727 = 0.007, where 16,727 bp is the length of the dog mitochondrial genome. Using the frequency of TCG triplets in the reference genome, we can calculate the expected number of T[C>T]G substitution types on the heavy strand as (total number of mutations) x (TCG frequency on the heavy strand) / 3 (as there are 3 possible C>N substitutions) i.e. expected number of T[C>T]G substitutions on the heavy strand = 835 x 0.007/ 3 $\approx$ 1.95. As we observed 22 T[C>T]G mutations on the heavy strand, the observed/expected ratio for this mutation type was 22/1.95 = 11.28.

Triplets within region MT:16129–16430 inclusive ('Substitution calling-Extraction and filtering'), as well as a set of specific excluded sites ('Additional quality checks and validation') were excluded from our analysis. This was accounted for during the calculation of expected substitutions described above.

## Selection analyses
### VAF
### Substitutions
Normalised VAF values for somatic substitutions were calculated as described in 'Host contamination'. Cumulative distribution functions for VAF scores (normalised to take into account host contamination, see 'Host contamination' above) were plotted for nonsense substitutions (n = 10) and missense and synonymous substitutions (n = 610), using the conservative somatic list (see 'Classification of tumour substitutions' above and *Supplementary file 4C*). Statistical significance was tested using the two-sample Kolmogorov-Smirnov test implemented in *R* (*R Development Core Team, 2013*).

### Indels
Normalised VAF for somatic indels was calculated as described in 'Predicted functional consequences on indels'. Cumulative distribution functions of normalised variant allele fractions were plotted for frameshift (n = 18) and non-frameshift (n = 9) indels. Statistical significance was tested using the two-sample Kolmogorov-Smirnov test implemented in *R* (*R Development Core Team, 2013*).

### dN/dS
dN/dS was estimated using a method adapted from *Martincorena et al. (2015)*. Briefly, a context-dependent model with 192 substitution rates (12 possible substitution types C>A, C>G, C>T, T>A, T>C, T>G, A>C, A>G, A>T, G>A, G>C, G>T x 4 possible 5' bases x 4 possible 3' bases) was used, thus accounting for any confounding context-dependent effects. The substitution rate was modelled as a Poisson process where the product of the underlying mutation rate and the impact of selection give the rate. A likelihood ratio test was used to test the deviation from neutrality ($w_{MIS}$=1 or $w_{NON}$=1), giving a *p*-value for the evidence of selection. To avoid any confounding effects due to the highly strand-biased CTVT mtDNA mutation spectrum (*Figure 2—figure supplement 1*), we excluded *ND6*, the only mitochondrially-encoded gene transcribed from the light strand, from this analysis.

## Acknowledgements
We thank Simon Frost, Young Seok Ju and Ludmil Alexandrov for helpful discussions and advice. This work was supported by grants from the Wellcome Trust (102942/Z/13/A) and the Royal Society (RG130615), and a Philip Leverhulme Prize from the Leverhulme Trust. We are grateful to Michael Stratton for his support. We would like to acknowledge the Core Sequencing Facility, IT groups and members of the Cancer Genome Group of the Wellcome Trust Sanger Institute. We are very grateful to Cathy King and all World Vets staff and volunteers who contributed to sample collection. We would like to thank the following individuals for useful information and for their help with obtaining samples for this project: Adrián Báez-Ortega, Ekaterina Batrakova, Rafaela Bortolotti Viéra, Matthew Breen, Austin Burt, Fernando Constantino Casas, John Cooper, Amici Cannis Cotacachi, Lytvynenko Dmytro, Ariberto Fassati, Ricardo Gaitan, David Hanzlíček, Mike Hobart, Rafael Ricardo Huppes, Matilde Jimenez-Coello, Debra Kamstock, Patrick Kelly, Tatiana Korytina, Ada Krupa, Anna

Kuznetsova, Olakunle AbdulRasaq Lawal, Thabo Lerotholi, Marco Lima-Maigua, Jimmy Loayza-Feijoo, Mayra López-Bucheli, Margarita Mancero-Albuja, Cynthia Marchiori Bueno, Talita Mariana Morata Raposo, Luis Martínez-López, Alfredo Martínez-Meza, Edward Migneco, Scott Moroff, Claudio Murgia, Alvira Murison Swartz, Fran Nargi, Edwin Ortiz-Rodríguez, Lisa Pellegrini, Gerry Polton, Freddy Proaño-Pérez, Clare Rebbeck, Šárka Rusá, Ceseltina Semedo, Ivan Stoikov, Lester Tapia, Mirela Tinucci Costa, David Walker, Robin Weiss, Kevin Xie, Yaping Zhang, veterinary workers of Pet Centre (UVAS, Lahore, Pakistan), staff at the WVS International Training Centre in Ooty (India), staff of Veterinary clinic "El Roble" and students from St. George's University (True Blue, Grenada, West Indies) who assisted with sample collection. We are grateful to the following organisations for helpful information: Associacao Bons Amigos de Cabo Verde, the Humane Society of Cozumel, ViDAS and Coco´s Animal Welfare, Veterinary Society of Surgical Oncology (VSSO), Veterinary Cancer Society, World Vets, Animal Balance, The Spanky Project, Humane Society Veterinary Medical Association-Rural Area Veterinary Services (HSVMA-RAVS), World Small Animal Veterinary Association (WSAVA), VWB/VSF Canada, Israel Veterinary Medical Association, Italian Veterinary Oncology Society, American College of Veterinary Internal Medicine (ACVIM), Rural Vets South Africa, Animal Care Association (The Gambia), МИР ВЕТЕРИНАРИИ (World Veterinary Medicine), and staff of VetPharma. We thank Ian Mickleburgh for his assistance with laboratory procedures. The map in *Figures 1C*, *3A* and *Figure 1—figure supplement 1* was used under the Royalty Free License from Free Vector Maps. Data associated with this study are available in GenBank with accession numbers: KU290400 - KU291095. Sequencing data associated with this study are available in the ENA with accession number ERP014691.

## Additional information

### Funding

| Funder | Grant reference number | Author |
| --- | --- | --- |
| Wellcome Trust | Investigator Award, 102942/Z/13/A | Elizabeth P Murchison |
| Leverhulme Trust | Philip Leverhulme Prize | Elizabeth P Murchison |
| Royal Society | Research Grant, RG130615 | Elizabeth P Murchison |

The funders had no role in study design, data collection and interpretation, or the decision to submit the work for publication.

### Author contributions

ASt, Critical reading of the manuscript, Acquisition of data, Analysis and interpretation of data, Drafting or revising the article, Contributed unpublished essential data or reagents; MNL, Critical reading of the manuscript, Acquisition of data, Analysis and interpretation of data, Drafting or revising the article; G-DW, T-TY, IA-O, JLA, KMA, LB-I, JLB, ACD, KFdC, AMC, HRC, JTC, SMCu, LDK, EMD, IAF, EFR, EF, SNF, FG-A, OG, RFHM, JJGPH, NI, DK, ML-P, RL, AMLQ, TL, GM, IM, SMCa, MFM-L, MM, BN, ABDN, WN, SJN, MMO, AO-P, MCP, RJP, JFR, JRG, HS, SKS, OS, RKS, AES-S, ASv, ITN, BAV, APdV, JPdV, OW, DCW, ASW-M, MGvdW, SAEW, Critical reading of the manuscript, Contributed unpublished essential data or reagents; EPM, Critical reading of the manuscript, Conception and design, Acquisition of data, Analysis and interpretation of data, Drafting or revising the article, Contributed unpublished essential data or reagents

### Author ORCIDs

Elizabeth P Murchison, http://orcid.org/0000-0001-7462-8907

## Additional files

### Supplementary files

• **Supplementary file 1.** Sample information. Summary of information available for 449 CTVT tumours and 338 hosts sequenced in this study. Includes data on location, year of collection, CTVT mtDNA clade, tumour and host mtDNA haplotypes, breed, age and sex.

• **Supplementary file 2.** Sequencing coverage and tumour cell fraction. (A) Average per-base coverage for whole genome (CanFam3.1) and for mtDNA genome (CanFam3.1; NC_002008). (B) List of 11 CTVT hosts with low coverage mtDNA regions. (C) Estimated tumour cell fraction for 449 CTVT tumours; tumour cell fraction was estimated by calculating the average VAF for variant substitutions present in tumour but not in matched host for each tumour.

• **Supplementary file 3.** Confirmation of CTVT diagnosis. Quantitative PCR (qPCR) was performed for *LINE-MYC,* a CTVT-specific rearrangement (Katzir et al., 1985; Katzir et al., 1987). Each reaction was performed in triplicate and a standard curve was used to detect relative DNA input at each locus. 'Normalised input' represents the relative *LINE-MYC* input detected in each sample normalised to *ACTB* ('Confirmation of canine transmissible venereal tumour (CTVT) diagnosis', Materials and methods). In general, we consider normalised input >0.05 as indicative of presence of *LINE-MYC.* Sufficient DNA was not available for samples 1380T and 1381T; diagnosis in these cases was performed with histopathology. *2T was grown as a xenograft.

• **Supplementary file 4.** Single point substitution variant lists. (A) Total number of substitution variants (n = 1005) identified in 449 CTVT tumours. (B) CTVT tumour somatic substitutions list (n = 928), including the average VAF value normalised for host contamination (see Materials and methods 'Classification of tumour substitutions'). Back mutations are not included on the list. (C) CTVT tumour conservative somatic substitutions list (n = 835), including the average VAF value normalised for host contamination (see Materials and methods 'Classification of tumour substitutions'). Back mutations are not included on the list. (D) Germline clade defining substitutions lists. Substitutions present in the pool of host substitutions and also shared between all samples within a clade (see Materials and methods 'Classification of tumour substitutions'). (E) Potential somatic substitutions lists. Substitutions not present in the pool of host substitutions, but shared between all samples within a clade (see Materials and methods 'Classification on tumour substitutions'). (F) Total number of substitution variants (n = 1152) identified in 338 CTVT host samples and 252 publicly available dog mitochondrial genomes (see *Supplementary file 8*).

• **Supplementary file 5.** Summary of small insertions and deletions (indels). (A) Total number of insertions and deletions identified in tumours (n = 27), including the average VAF value normalised for host contamination (see Materials and methods) (B) Total number of homoplasmic insertions and deletions in CTVT hosts (n = 7), including the average VAF value (see Materials and methods).

• **Supplementary file 6.** Annotation of single point substitutions. Annotation of individual point substitution mutations in (A) 449 CTVT tumours (see list *Supplementary file 4A*, excluding back mutations) and (B) 338 CTVT hosts (see list *Supplementary file 4F*). Annotation was performed using Variant Effect Predictor (McLaren et al., 2010). In cases where a single substitution affects two different genes, the two annotations are shown on different lines.

• **Supplementary file 7.** Annotation of insertions and deletions (indels). Annotation of individual indels (A) unique to CTVT tumours and (B) homoplasmic in CTVT hosts. Annotation was performed using Variant Effect Predictor (McLaren et al., 2010).

• **Supplementary file 8.** Publicly available mitochondrial dog genomes used in the study. Summary of Genbank accession numbers and metadata for 252 publicly available dog mitochondrial genomes

included in this study (see Materials and methods 'Extracting substitution variants from publicly available dog sequences' and *Supplementary file 4F*).

• Supplementary file 9. Timing analysis. Methods used to estimate the time since the origin of CTVT clades 1 to 5. Potential somatic substitutions (see Materials and methods 'Classification of tumour substitutions' and *Supplementary file 4E*) are those which are shared between all tumours within a clade, but that are not found in the normal dog population; thus we cannot confirm their germline or early somatic status. The upper panel lists time estimates (years before present for the origin of each clade) assuming potential somatic substitutions are somatic and arose after mtDNA horizontal transfer; the lower panel lists time estimates (years before present for the origin of each clade) assuming potential somatic substitutions are germline and were originally present on the mtDNA haplotype that founded the clade.

• Supplementary file 10. Summary of back mutations. List of back mutations attributable and non-attributable to a putative recombination event.

• Supplementary file 11. CTVT tumour and host mtDNA haplotype lists.

• Supplementary file 12. Substitutions with corresponding VAF (before normalisation) for each of 449 CTVT tumours and 338 CTVT hosts. *Supplementary file 12* is included as accompanying zip file.

• Supplementary file 13. Indels with corresponding VAF (before normalisation) for each of 438 CTVT tumours and 334 CTVT hosts. Samples with very high coverage of the mitochondrial genome were excluded from the indel analysis (see Materials and methods 'Indel calling-Extraction and filtering'). *Supplementary file 13* is included as accompanying zip file.

### Major datasets

The following datasets were generated:

| Author(s) | Year | Dataset title | Dataset URL | Database, license, and accessibility information |
|---|---|---|---|---|
| Strakova et al | 2016 | Mitochondrial genetic diversity, selection and recombination in a canine transmissible cancer | http://www.ncbi.nlm.nih.gov/genbank/KU290400 | Publicly available at Genbank (accession no: KU290400 - KU291095) |
| Strakova et al | 2016 | Canine transmissible venereal tumour (CTVT) mitochondria | http://www.ebi.ac.uk/ena/ERP014691 | Publicly available at European Nucleotide Archive (accession no: ERP014691) |

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
