## [Decision Letter]

Thank you for submitting your article "Mitochondrial genetic diversity, selection and recombination in a canine transmissible cancer" for consideration by *eLife*. Your article has been reviewed by three peer reviewers, and the evaluation has been overseen by Elaine Ostrander as the Reviewing Editor and Mark McCarthy as the Senior Editor.

The reviewers have discussed the reviews with one another and the Reviewing Editor has drafted this decision to help you prepare a revised submission.

Summary:

This is a well-written and concise paper that addressed the next natural step in unraveling the mystery of CTVT in the dog. The figures are clear (and attractive). Overall the paper does an outstanding job documenting, through low-coverage sequencing and phylogenetic analysis, the frequent transfer of mitochondrial DNA from the affected host to the canine transmissible venereal tumor (CTVT). Sequencing reveals that acquisition of host mtDNA has occurred at least five times during the worldwide spread of the clonal tumor, and that these five clades acquired their mtDNAs around 1,000 years ago. The sequences further suggest that there has been retention of ORFs, suggesting selection for function in the acquired DNAs. The data for the five clade hypothesis and horizontal transfer is convincing. The paper can be strengthened however in a few ways.

Essential revisions:

The reviewers felt this was a strong and important contribution to an interesting field. They identified several issues, however, that need to be addressed. These include:

1) Two reviewers raised the issue of the Indian-CTVT clade (IV), which implies highly non-clock-like evolutionary rates within the ancestor of the lineage. It is found at the end of fairly long branches and it may not be accurately placed within the phylogeny. One possibility is that this observation may be the result of the ML search that was performed [(Phyml with rudimentary branch swapping – a "combination of NNI and SPR")]. This approach may be insufficient to search treespace with such a large number of OTUs. The reviewers ask that the analysis be repeated with the newest version of RAxML which has more thorough search options. Alternatively, the authors should closely check the sequences or alignment as this could be due to minor alignment errors.

2) Two reviewers also raised the issue of the full-length mitochondrial NUMT that is found in the dog genome. The reviewers state that NuMTs are a major confounder in many species, and it was well-served to be addressed. However, over 150 NuMTs have been discovered in the canine genome. It is unclear how the wgsim recreation of reduced CTVT reads aligned to the canine genome would demonstrate how NuMTs are unlikely to contribute significantly to mtDNA variant calling. During CTVT evolution, particularly in an organism purported to engage in mictochondrial capture, it is likely that novel NuMTs were introduced that diverge from the canine genome. While the authors are likely correct that current mtDNA assemblies have not been compromised by NuMTs, if the NuMT is tandemly arrayed (as they are in many mammalian genomes) it could be highly multicopy and approach the read depth of the true cytoplasmic copies. The authors should check this by estimating read depth of the dog NuMT insertion within the nuclear genome and see if is collapsed and determine the expected depth based on this measurement, rather than assuming one copy. This will certainly raise the caliber of the paper.

3) The time estimate of 11,000 years cited in this work was estimated using a subset of extant canine variation to isolate ~1.9 million somatic mutations, and perform a strict clock approach based on the mutation signature of human medulloblastoma. Subsequent work used a more comprehensive catalog of canine variation resulting in a 50% reduction in total somatic mutations. A reanalysis of the timing of CTVT origin may be warranted and particularly useful for the field.

4) The authors discuss the phenomenon of mitochondrial recombination, but do not discuss the likelihood of heteroplasmy. If mitocapture is a mechanism prevalent in CTVT, it is likely that these tumors are heteroplasmic. Are the authors indicating that mitochondria from the original CTVT founder, or a subsequent host prior to Clade A is no longer present, and has been supplanted completely by now homoplastic horizontal transfer? The issue of tumor clonality / subclonality is not addressed.

5) The authors use an in-house variant caller that is based on paired tumor-normal samples. Since CTVT by its nature does not have a normal sample, why was another caller that did not rely on tumor /simulated normal used? Please explain.

6) Why was ploidy estimated to be 1.5 for CTVT and 2.0 for host? The reference indicates that CTVT is diploid.

7) Importantly, there is no indication of where the mtDNA alignments or variants, and low coverage nuclear data is deposited for future replication. This must be done *prior* to acceptance of the paper for publication.

---

## [Author Response]

Essential revisions:

*The reviewers felt this was a strong and important contribution to an interesting field. They identified several issues, however, that need to be addressed. These include: 1) Two reviewers raised the issue of the Indian-CTVT clade (IV), which implies highly non-clock-like evolutionary rates within the ancestor of the lineage. It is found at the end of fairly long branches and it may not be accurately placed within the phylogeny. One possibility is that this observation may be the result of the ML search that was performed [(Phyml with rudimentary branch swapping – a "combination of NNI and SPR")]. This approach may be insufficient to search treespace with such a large number of OTUs. The reviewers ask that the analysis be repeated with the newest version of RAxML which has more thorough search options. Alternatively, the authors should closely check the sequences or alignment as this could be due to minor alignment errors.*We thank the reviewers for their comments regarding the phylogenetic position of CTVT clade 4 tumours in Figure 1, and for raising the concern that this group may be inaccurately positioned due to limitations in the phylogenetic analysis algorithm or errors in alignment.

We repeated the phylogenetic analysis shown in Figure 1 using RAxML 8.2.6 using the default rapid hill climbing algorithm. The phylogenetic position of the clade 4 tumours was consistent between the PhyML and RAxML trees (see Figure 4). We have now included a statement in the Methods section: “Phylogenetic trees constructed using RAxML 8.2.6 with a rapid climbing algorithm and identical model specification were consistent with the phylogenetic trees obtained using PhyML.”

Author response image 1.Diagram showing phylogenetic trees constructed using PhyML and RAxML.**DOI:**
http://dx.doi.org/10.7554/eLife.14552.028

In addition, we performed manual validation of each of the variants identified in CTVT clade 4 tumours, and visualised the alignment of clade 4 haplotypes to confirm the absence of alignment errors. We have added the following statement to the Methods section: “The long branch-lengths in CTVT clade 4 were further investigated by visually validating sequence alignments involving these samples.”

We would further like to clarify that we do not wish to imply that clade 4 mtDNA represents the “ancestor of the lineage”. Our interpretation of the data is that the clade 4 group is derived from one of the five distinct horizontal transfer events detected in our analysis; we do not believe that the clade 4 mtDNA was originally present within the founder CTVT dog.

2) Two reviewers also raised the issue of the full-length mitochondrial NUMT that is found in the dog genome. The reviewers state that NuMTs are a major confounder in many species, and it was well-served to be addressed. However, over 150 NuMTs have been discovered in the canine genome. It is unclear how the wgsim recreation of reduced CTVT reads aligned to the canine genome would demonstrate how NuMTs are unlikely to contribute significantly to mtDNA variant calling. During CTVT evolution, particularly in an organism purported to engage in mictochondrial capture, it is likely that novel NuMTs were introduced that diverge from the canine genome. While the authors are likely correct that current mtDNA assemblies have not been compromised by NuMTs, if the NuMT is tandemly arrayed (as they are in many mammalian genomes) it could be highly multicopy and approach the read depth of the true cytoplasmic copies. The authors should check this by estimating read depth of the dog NuMT insertion within the nuclear genome and see if is collapsed and determine the expected depth based on this measurement, rather than assuming one copy. This will certainly raise the caliber of the paper.

We appreciate the reviewers’ comment about nuclear copies of mitochondrial DNA (NuMTs). Further to the concern raised, we have expanded the Methods subsection entitled “Nuclear copies of mtDNA (NuMT) analysis” to clarify the rationale behind the NuMT analysis that we have performed and to provide further justification for our conclusion that NuMTs have not significantly impacted on our CTVT mtDNA variant catalogue.

*3) The time estimate of 11,000 years cited in this work was estimated using a subset of extant canine variation to isolate ~1.9 million somatic mutations, and perform a strict clock approach based on the mutation signature of human medulloblastoma. Subsequent work used a more comprehensive catalog of canine variation resulting in a 50% reduction in total somatic mutations. A reanalysis of the timing of CTVT origin may be warranted and particularly useful for the field.*

We thank the reviewers for raising this point regarding the timing of CTVT origin. We agree that future re-analysis of the timing of CTVT’s origin will be an important contribution to the field. However, we believe that re-analysis of the timing of CTVT origin is beyond the scope of this work, whose focus is on mtDNA evolution in CTVT (and as mtDNA in CTVT has been acquired by horizontal transfer, analysis of CTVT mtDNA cannot cast light on the timing of CTVT’s origin). Nevertheless, we are aware of the uncertainty in CTVT origin timing estimates, and have emphasised this uncertainty by stating “approximately” or “around” whenever 11,000 years is mentioned. Additionally, we have now cited the paper to which the reviewers refer (Decker et al., 2015) (see the Introduction).

*4) The authors discuss the phenomenon of mitochondrial recombination, but do not discuss the likelihood of heteroplasmy. If mitocapture is a mechanism prevalent in CTVT, it is likely that these tumors are heteroplasmic. Are the authors indicating that mitochondria from the original CTVT founder, or a subsequent host prior to Clade A is no longer present, and has been supplanted completely by now homoplastic horizontal transfer? The issue of tumor clonality / subclonality is not addressed.*We thank the reviewers for raising this interesting point. As the reviewers suggest, we do believe that the absence of haplotype-level heteroplasmy suggests that mtDNA horizontal transfer is either (i) not common or (ii) does not frequently reach levels that are detectable in our analysis. Following from this, we are indeed suggesting that the original CTVT mtDNA haplotype has been replaced and is no longer present, as stated in the third paragraph of the Results and Discussion section.

Following on from this, we would like to thank the reviewers for the very valid point regarding tumour subclonality. The variable VAFs reported in Figure 2 may well be due to both intracellular heteroplasmy as well as intercellular heterogeneity (subclonality). We have addressed this point by adding a sentence to the Methods section, reading “Tumour variants with normalised VAF<1 most likely represent heteroplasmic variants; however, we cannot exclude that these represent distinct cellular subclones harbouring distinct homoplasmic mtDNA populations”.

*5) The authors use an in-house variant caller that is based on paired tumor-normal samples. Since CTVT by its nature does not have a normal sample, why was another caller that did not rely on tumor /simulated normal used? Please explain.*We appreciate the reviewers’ concerns about the variant caller that we used in our analysis. The unavailability of matched normal DNA from the original founder dog that first gave rise to CTVT poses challenges for the computational analysis of CTVT. Somatic variant callers, which usually require paired tumour-normal samples, generally have greater sensitivity for variant detection and do not invoke underlying diploid segregation models. We selected CaVEMan as a variant caller, as this algorithm is widely used and validated as a somatic variant caller.

*6) Why was ploidy estimated to be 1.5 for CTVT and 2.0 for host? The reference indicates that CTVT is diploid.*We are very grateful to the reviewers for pointing out this error. The reference indeed indicates that CTVT is diploid and we have now corrected the calculations to reflect this (please see Results and discussion, first paragraph and subsection “DNA sequencing”, first paragraph).

7) Importantly, there is no indication of where the mtDNA alignments or variants, and low coverage nuclear data is deposited for future replication. This must be done prior to acceptance of the paper for publication.

Further information about deposition of the data associated with this study is detailed below:

a) The mtDNA data associated with this study are available in GenBank with accession numbers: KU290400 – KU291095.

b) All the variant lists are available as Supplementary datasets 1 and 2.

c) The low coverage sequence data associated with this study are available in the ENA with accession number ERP014691.